# A Neural Network Approach to Estimate the Frequency of a Cantilever Beam with Random Multiple Damages

**DOI:** 10.3390/s23187867

**Published:** 2023-09-13

**Authors:** Prattasha Saha, Mijia Yang

**Affiliations:** Department of Civil, Construction and Environmental Engineering, North Dakota State University, Fargo, ND 58104, USA; prattasha.saha@ndsu.edu

**Keywords:** multiple damage, natural frequency, finite element method, ANN, linear regression

## Abstract

Natural frequency is an important parameter in the structural health monitoring (SHM) system. Any changes in this parameter indicate structural alteration due to damage. This study provides a neural network (NN) solution as an alternative to the finite element (FE) method to measure the natural frequencies of a cantilever beam with random multiple damage. It is based on a statistical dataset of a free vibration test obtained from the APDL (Ansys parametric design language) simulation using a MATLAB (matrix laboratory) script. The script can generate an unlimited number of possible damage combinations for any given parameters with the help of the Monte Carlo (MC) technique. MC helps to generate a random number of damages in random locations at each simulation. Damage conditions are controlled by three parameters including damage severity and damage size (in terms of the mean and standard deviation of damage). Moreover, the method proposes a curve-fitting equation to validate the predicted natural frequency for the first three modes obtained from the neural network model. Both methods are in good agreement with each other, having minimal errors in the range of 0.2–3% for each mode. The frequency result shows that the beam frequency is 8.6486 Hz if the area reduction is 10%, whereas it comes down to 7.2338 Hz if there is a 30% area reduction. A two-level factorial test shows that damage severity is the most impactful factor compared to the damage sizes on the frequency shift event. This indicates that damage alters the composition of the beam and has an impact on its frequency change with the assumed damage parameters. Therefore, the proposed NN model can estimate the frequency shift for various damage scenarios. It can be utilized in the vibration-based damage identification process to predict the frequency changes of the damaged beam without any computational burden.

## 1. Introduction

Geometric shape, mass, stiffness, and related boundary conditions can alter natural frequencies and corresponding mode shapes to define structural dynamic behavior [1,2]. The structural system fails either suddenly or gradually when it is associated with different types of damage. The damage identification process provides key information about the status of a structure to verify its structural assurance [3]. Non-destructive testing like visual inspection, an ultrasound, and a magnetic field test can be performed to detect the damages or defects in structural elements, though these are not applicable to inaccessible members of a structure [4,5]. However, natural frequencies and mode shapes can be the parameter to assess structural damage because damage changes its vibration characteristics [6,7,8]. Regarding this inverse problem of damage investigation, the natural frequency calculation of a damaged beam plays an important role. For an undamaged beam, natural frequency can be calculated theoretically. But damage can affect its global or local stiffness, material characteristics, as well as its dynamic parameters [9,10,11], thus making it difficult to theoretically predict the frequencies of damaged elements. The paper will review the current literature on this topic in the introduction section. A detailed theoretical background on microstructure generation, APDL numerical modeling, neural network, and statistical analysis will be included in Section 2. Validation and case studies will be included in Section 3. Result analysis and conclusion will follow in Section 4 and Section 5, respectively.

Prior research has been conducted to identify structural damage by utilizing natural frequencies. Kumar et al. developed a high-order kinematic model and used the virtual crack closure technique for measuring natural frequencies. They utilized the vibration responses to investigate the damaged shell structure [12]. Dubey et al. used an algorithm based on the frequency shift coefficient to estimate both the damage severity and position [13]. Their modal parameter calculation is based on solving the eigen values obtained from a finite element model. Reynders et al. [14] proposed a local flexibility method for vibration-based damage localization and quantification which is applicable for single and multiple cracks. The resulting accuracy is sensitive to the number of modes that vary with the change of damage locations and their severities. Caddemi et al. [15] derived a closed-form solution to evaluate the frequencies and the vibration modes of multi-cracked frame structures. It solved the exact dynamic stiffness matrix of concentrated damages. Kim et al. [16] utilized two mode shapes and natural frequencies for their damage index, damage sizing, and damage localization algorithms to identify damages. These approaches lost their accuracy in the quarter span area.

According to recent studies, machine learning (ML) approaches have been utilized to develop a wide array of vibration-based damage detection through supervised and unsupervised algorithms [17,18]. Samali et al. [19] proposed frequency response functions (FRFs) along with an artificial neural network (ANN) to locate damage locations and severities of notch-type damages in a framed structure. They noticed that the FRF data showed less sensitivity toward local modes. Bandara et al. [20] also used FRF data of undamaged and damaged structure along with an ANN and a principle component analysis (PCA) to quantify damage scenarios. This combined method can work with noisy data without an eigenvalue analysis or any optimization. Ho et al. developed a coupled model between the ANN and antlion optimizer method based on the stochastic theory to locate the damage position. They used the FE model to obtain modal parameters and then generated a damage index as the main damage localization factor [21]. The ML-based surrogate model along with natural frequencies, mode shapes, and damage indexes is proposed by Lee et al. [2], where-in due to limited sensors, displaced mode shapes were not available for all of the degrees of freedom. Therefore, to reduce the neural network size, they used a modal strain energy-based index. They trained their neural network model with natural frequencies obtained from a modal analysis. Hakim et al. did an experimental setup to extract frequency data from the modal estimation method [22]. Their proposed ANN based model can successfully predict the location and severity of small-scale damage in fixed–fixed support conditions.

However, the connection between damage morphology and the frequency shift is still elusive. There is no robust tool that can create the relationship between damage configuration and frequency changes. In this paper, the authors will resort to the Monte-Carlo microstructure generation, APDL advanced numerical simulation, and neural network (NN) model and statistical analysis to capture such a correlation. The goal of the research is to develop an ANN-based model to predict the frequency of a damaged beam and compare the obtained frequency shift result with a statistical fitting equation. The assumed damage conditions for this study vary with the change of damage extent, size, and location. The proposed NN model proposed will be an efficient tool for frequency estimation according to different states of multiple damages occurring in a beam structure. Though environmental conditions like temperature changes, wind, humidity, or other external factors also have impacts on vibration properties [23], only damage uncertainties are considered here.

## 2. Methodology

Damage in a structure alters its structural stiffness along with a change in modal parameters, from which we can measure the natural frequencies. There is a deterministic relationship between stiffness change and the decrease in the natural frequencies of a damaged structure. Hence, a finite element model is developed to analyze the frequency shift of a damaged cantilever beam under a free vibration condition.

### 2.1. Microstructure Generation

The Monte Carlo (MC) method is an important tool to generate random numbers for solving analytical problems. It also provides the convergent result for an unknown exact value [24,25]. In this paper, a beam with multiple damages is simulated in MATLAB R2021a software using the MC concept. The generated circular damages throughout the beam length follow a normal distribution. During each simulation, damage location as well as the number of damages changes corresponding to the input variables like the mean and standard deviation of the damage size and its severity. The normal distribution (or bell curve) formula is [25]
(1)$$f\left(x\right)=\frac{1}{\sigma \sqrt{2\pi \;}}\;\;e^{\frac{-{\left(x-\mu \right)}^{2}}{{2\sigma }^{2}}}\;\;$$

In Equation (1), σ and μ are the mean and standard deviation (SD) of damage size, respectively, π = 3.1416, and *e* = 2.71828; *x* = distance in the horizontal axis. The mean gives the average value of the damage size and the SD is a summary measure of the differences between each damage from the mean. These are useful measures of any scattered damage. In this distribution, a range covered by 1 SD above the mean and 1 SD below includes about 68% of the observations; a range of 2 SD above and 2 SD below covers about 95% of the observations; and 3 SD above and 3 SD below covers about 99.7% of the observations [26]. Here, damage sizes are followed by both the mean and SD values. On the other hand, the bending moment controls the effect of the damage position. Damage does not cause frequency degradation when it is located at an inflection point but damages on maxima will affect it mostly [27]. Considering this concept, for a beam with multiple damages having a damage severity of 10–30%, the mean values ranging from (2 to 4) m and (0.1–0.3) m SD values are used to generate different damage scenarios. These three input variables including damage severity (λ), mean (α), and SD (µ) control the damage size and its location by following the Monte Carlo formula to generate each damage condition. The ranges of the parameters are selected accordingly to avoid computational error. Here, the damage severity is defined as the percentage of area damaged (voids) compared to the total area. A greater damage severity will highly impact the structural stiffness or other modal parameters and eventually makes the frequency estimation more difficult. Thus, a higher percentage of severity (>30%) and a very small percentage pose challenges to the meshing of the FE model. For this reason, the proposed model is limited to a range of damage severity and its size.

### 2.2. Frequency Extraction

Figure 1a,b shows the damaged beam according to the given α, µ, and λ. Here, the random multiple damages are generated throughout the beam by using the Monte Carlo method in MATLAB.

In this study, a finite element model is proposed to study the frequency shift of a cantilever beam under the free vibration condition. The numerical model is developed using ANSYS Mechanical APDL (a commercially available software). Though the frequency result is extracted from APDL, the beam is at first simulated with multiple random damages through a MATLAB script. Figure 1 shows an image of the damaged beam with multiple random damages generated from the MATLAB script.

For the simulation purpose, the dimension of the beam (Figure 2) is considered as 100 m × 10 m × 1 m (= length (L) × width (b) × depth (h)). To perform a free vibration test, the beam is fixed at one end and free at another end (acting as a cantilever beam). The corresponding material properties are given in Table 1. For the simulation, the beam is selected as a solid (quad 4 node 182) type having elastic isotropic properties.

The MATLAB script can be directly imported in APDL to generate natural frequencies in the first three modes along with its undeformed or deformed shape of each mode (Figure 3a–d). Only the input variables need to be changed in the script for a new damaged condition. This approach reduces computational time in an efficient way. For further analysis, a frequency table is generated for the first three modes. The table consists of 45 datapoints individually for all the three modes. Each datapoint has an average value of three frequencies generated from three random simulations for each damage condition.

Mesh sensitivity is critical in the FE modeling results. In this study, the desired mesh size is determined after a mesh sensitivity analysis. The frequency change of the first three modes of the studied beam with mesh sizes is given in Figure 4a–c. It shows that for mode 1, 2, and 3, the frequency of a damaged beam comes to an equilibrium state when the mesh element size is reduced to less than 1.0 m. Considering the sensitivity analysis result and the arbitrary sizes of damages, the element edge length is chosen as 0.1 m. Figure 4d presents the meshed elements in detail near a damaged location.

### 2.3. Machine Learning

The artificial neural network (ANN) has the ability to build a relationship between structural and modal parameters through a training process [28]. It is also a convenient tool for complex and non-linear problems [18]. It can predict non-linearities of n-dimensional input values. A general NN model consists of several neurons connected with each other and organized in several layers (Figure 5). A non-linear function helps the network to process the information that it receives from its connections. The topology of feed forward neural networks includes information flowing from one layer to another through the network until it reaches desirable outputs [29,30].

In this paper, numerically obtained frequency results of the first three modes of damaged beams are used as the required dataset to train the ANN model. The dataset generated from the FE model contains 45 damage scenarios for three parameters—damage severity (λ), mean (α), and SD (µ) of damage size. It is randomly divided into three segments including training, testing, and the validation set. About 70% of the data is used for training, 15% for testing, and 15% for the validation. During training, the values of three damage parameters are used as the input data and the output is the natural frequencies in three modes corresponding to each damage condition.

The model is developed using the neural net fitting tool (nftool) in MATLAB software. It utilizes a two-layer feed-forward network with sigmoid hidden neurons and linear output neurons. It is trained with the Levenberg–Marquardt backpropagation algorithm to obtain network weights using the mean square error function. The Tansig transfer function is used to calculate the output from its net input. The model performance obtained from MATLAB is shown in Figure 6. The result indicates that R^2^ values of training, validation, and the testing set are above 99%. The training dataset has 45 datapoints and 3 input parameters. Thus, the hidden layers and the hidden neurons cannot be too many in the NN model; otherwise, it will cause an overfitting problem. Through trial and error, the model is trained with one hidden layer, having 2 to 9 neurons. To obtain the optimized geometry, a model with 5 hidden neurons is proposed. The selected number of hidden neurons in the hidden layer gives the best performance to predict frequency. Figure 7 shows the optimal ANN model structure.

### 2.4. Statistical Approach

To perform damage detection through the frequency changes of beam structures, it is generally composed of two phases—the forward problem and the inverse problem [31]. In the forward problem, frequencies of a known damage scenario are calculated, which are used later on to calculate the damage parameters in the inverse problem [32]. In this case, the accuracy of the finite element model (FEM) and its measured frequencies gets reduced due to some uncertainties, such as a lack of boundary conditions, or structural nonlinear behavior [33]. Hence, a statistical formula has been developed to reduce the computational drawbacks of the FE method as well as the tool to validate the proposed NN model. For a healthy beam with a uniform cross-sectional area, we can obtain its natural frequencies for each mode by using the analytical formula. The governing equation to measure the natural frequency of a healthy or undamaged cantilever beam during a free vibration test is
(2)$$f=(\frac{1}{2\pi })\cdot k_{n}^{2}\sqrt{\frac{EI}{\rho AL^{4}}}\;\;\;$$
(3)$$I=\frac{1}{12}bh^{3}\;$$

Here, *E* = Youngs modulus, *I* = Moment of inertia, ρ = density, *A* = cross sectional area, *L* = beam length, *b* = beam width, *h* = beam depth, and *n* = mode number. Wherein, for the 1st mode, *k* = 1.875; for the 2nd mode, *k* = 4.694; and for the 3rd mode, *k* = 7.855. For error analysis,
(4)$$e=(\frac{f-f^{\prime }}{f^{\prime }})\cdot \;100\%$$

Here,

f = Natural frequency obtained from the numerical simulation; and

$f^{\prime }$ = Natural frequency obtained from analytical calculations.

But the natural frequencies of a beam with multiple damages cannot be measured by Equation (2) due to its non-uniform cross-sectional area. Therefore, a frequency dataset generated from the FEM simulation using the three input damage parameters including damage severity (λ), damage size and distribution in terms of the mean (α) and standard deviation (µ), respectively, has been used to solve this problem analytically. The dataset was generated using damage severity (λ) ranging from 10% to 30% with different damage mean sizes (α) and SDs (µ). The range for both α and µ is mentioned in Section 2.1. A 3rd order polynomial function is used to fit the first three mode frequencies of the damaged beam for any given damage parameter.
(5)$$\begin{array}{l} F_{predicted}=A\alpha +B\mu +C\lambda +D\alpha^{2}+E\mu^{2}+f\lambda^{2}+g\alpha^{3}+H\mu^{3}+I\lambda^{3}+J\alpha^{2}\mu +K\alpha \mu^{2}+L\alpha \lambda^{2}+M\lambda \alpha^{2}+N\mu \lambda^{2}+\\ O\mu^{2}\lambda +P\alpha \mu +Q\mu \lambda +R\lambda \alpha +S\alpha \mu \lambda \end{array}$$

In Equation (5), A to S are the coefficients. We used the least square (LS) linear regression method to determine the coefficients by using Equation (6). After that, these coefficient values are used in Equation (5) to calculate the predicted natural frequencies. A good least-squares fit is indicated by a large value of r, the maximum value of which is 1. To check qualitatively how good the regression is, the r value (co-relation factor) was extracted from Equations (7) and (8). The flowchart of this statistical approach is shown in Figure 8.
(6)$$\begin{array}{l} F_{actual}=A\sum_{i=1}^{n}\alpha (i)+B\sum_{j=1}^{n}\mu (j)+C\sum_{k=1}^{n}\lambda (k)+D\sum_{i=1}^{n}{\alpha (i)}^{2}+E\sum_{j=1}^{n}{\mu (j)}^{2}+f\sum_{k=1}^{n}{\lambda (k)}^{2}+g\sum_{i=1}^{n}\alpha {\left(i\right)}^{3}+\\ H\sum_{j=1}^{n}{\mu (j)}^{3}+I\sum_{k=1}^{n}{\lambda (k)}^{3}+J\sum_{i=1,j=1}^{n,n}{\alpha (i)}^{2}\mu (j)+K\sum_{i=1,j=1}^{n,n}\alpha \left(i\right)\mu {\left(j\right)}^{2}+L\sum_{i=1,k=1}^{n,n}\alpha \left(i\right)\lambda {\left(k\right)}^{2}+\\ M\sum_{i=1,k=1}^{n,n}\lambda (k)\alpha {\left(i\right)}^{2}+N\sum_{j=1,k=1}^{n,n}\mu (j)\lambda {\left(k\right)}^{2}+O\sum_{j=1,k=1}^{n,n}\mu {\left(j\right)}^{2}\lambda (k)+P\sum_{i=1,j=1}^{n,n}\alpha \left(i\right)\mu (j)+Q\sum_{j=1,k=1}^{n,n}\mu (j)\lambda (k)+\\ R\sum_{i=1,k=1}^{n,n}\lambda (k)\alpha (i)+S\sum_{i=1,j=1,k=1}^{n,n,n}\alpha \left(i\right)\mu (j)\lambda (k) \end{array}$$
(7)$$r^{2}=1-\frac{s}{s_{0}}\;\;$$
where,
(8)$$s={\left(F_{predicted}-F_{actual}\right)}^{2}\;\mathrm{and}\;s_{o}={\left(F_{actual}-F_{average}\right)}^{2}$$

### 2.5. Data Analysis

To analyze the effect of three variables—the mean, standard deviation, and damage severity (α, µ, and λ, respectively)—on the frequency shift obtained from the statistical regression (Equation (5)), a two-level factorial test [34] is performed. Table 2 presents randomly-selected two-level (low (−) and high (+)) values of each variable that are used for this test. As a two-level, three-factor (α, µ, and λ) design, the analysis utilizes eight of the obtained frequencies from Equation (5) based on the combinations of α, µ, and λ specified in Table 2. The effects of the variables are calculated from Equation (9) using eight test conditions.
(9)$$Effect=\frac{\sum Y_{+}}{n_{+}}-\frac{\sum Y_{-}}{n_{-}}$$

Here, *n*_+_ and *n*_−_ are the number of datapoints at each level; *Y*_+_ and *Y*_−_ are the associated frequency value. To calculate the effect of variables on the frequency shift (Equation (9)), variables are defined whether they are in the low or high level.

## 3. Model Validation

### 3.1. Undamaged Beam

Since Equation (2) is applicable for a uniform cross-sectional area, the natural frequencies of the undamaged beam that are extracted from the proposed FE model can be validated with it. The result (Table 3) shows that frequencies from the FE simulation and theoretical are in good agreement, having a relative error of 0.01–12% from mode 1 to mode 3, respectively.

### 3.2. Damaged Beam

To calculate the damaged beam frequency, the proposed ANN model is compared with the polynomial functions (Equation (5)). Since damage changes the structural stiffness, the typical analytical solution (Equation (2)) cannot be applied to measure its vibration response. A new frequency dataset containing three damage cases obtained from the FE model has been used to test the trained neural network model. The ANN model can successfully predict the frequency for the first three modes which shows a similar trend with the FE results as well as the statistical solution (Figure 9).

## 4. Result and Discussion

For damage identification in a forward problem, frequency change is a good feature since it is less noise-contaminated [35]. This paper addresses a neural network model to predict the natural frequency of a damaged cantilever beam. The damaged condition is indicated by different levels of damage severity with a random size and random location of multiple damages. The proposed neural network model is trained with a frequency dataset which is obtained from an FE model. A MATLAB script is programed to generate the APDL version of the FE model, and this process reduces both the computational cost and time of the numerical simulation. The Monte Carlo method is used to generate multiple damages in a random location throughout the beam length. Every FE simulation presents a new pattern of damage scenario throughout the beam length with a change in damage size and location. Even if the distribution parameter remains the same, the damage pattern also keeps changing for every simulation. The obtained frequency dataset from the FE model shows that the frequency level increases gradually from a lower mode to a higher mode. This trend is applicable for any kind of damage severity, any damage location, and for any damage size.

After the training process, the artificial neural network (ANN) model can successfully predict the natural frequency for any given λ,α,µ value. The undamaged and healthy beam frequency from the proposed FE model is almost the same as the theoretical result with a maximum error rate of around 12%. Table 3 shows that in both results, frequency is always greater in a higher mode compared to the immediate lower one. Theoretically, the frequency in the first mode is 0.8381 Hz which gradually increases to 14.822 Hz in the third mode. Similarly, in the FE result, the 0.838 Hz in the first mode increased to 12.989 Hz while it comes to the third mode. On the other hand, frequency for a damaged beam cannot be obtained by using Equation (2) since the beam is not in a uniform cross-sectional area because of damage. Due to this reason, the NN model is proposed. The R value from the regression plot (Figure 6) shows that the model performance is well enough regarding the correlation between outputs and targets. The predicted natural frequency of the damaged beam from the proposed ANN model needs an alternative way for its validation. Thus, a third order polynomial equation is developed to predict the natural frequency.

To validate the ANN model, the statistical third order polynomial solution has been developed using a linear regression model. Though the frequency shift follows a nonlinear relation with the change of damage severity and the damage size and location, the coefficient values are almost similar whether we use a linear or nonlinear curve fitting method. To avoid complexity, the linear regression model is applied here. To check the performance of the regression model, the co-relation coefficient (r) is also extracted. The value of the co-relation coefficient in mode 1, 2, and 3 is 0.85, 0.86, and 0.85 respectively, which means the regression result is acceptable.

Then, three new damage conditions (in terms of damage severity, damage size, and location) are used to generate a new dataset. This new dataset helps to compare the frequency prediction of the trained ANN model with the FE result. The comparison indicates that both the ANN and statistical approach show a similar trend to predict the natural frequencies of the damaged beam for the given damage conditions. Moreover, they are also in good agreement with the FE result (Figure 9). Table 4, Table 5 and Table 6 show the comparison of the predicted frequency shifts for mode-1, mode-2, and mode-3 obtained from the FE model, ANN, and statistical fitted equation using the new damage scenarios. As shown by the ANN model (Table 4), when the damage severity increases from 10% to 30%, the natural frequency decreases because structural elements lose their stiffness due to damage. In the first mode (Table 4), the model predicts the frequency as 0.7958 Hz under 10% of damage extent, which comes to 0.7940 Hz after increasing the damage to 30%. The same trend is followed by all the three modes as well as the statistical result. Table 4, Table 5 and Table 6 show that in every mode, for 20% damage along with other variables, the frequency shift is off the trend, which is larger than 30% damage but also greater than 10% damage. For example, according to the ANN model, in mode 2 and 3, frequency is increased by 0.26 Hz and 1.42 Hz, respectively, with the increase of damage extent from 10% to 30%, whereas this frequency rate drops at 20% damage severity. This happens because the frequency shift depends on the damage extent as well as on the damage size and location. This change (Table 4, Table 5 and Table 6) happens not only with respect to damage severity but also with the change of the damage size. The frequency change can be similar or different in some cases if damage occurs in different severity and in different locations with multiple cracks [35]. If the maximum damage location is near to the fixed end, it makes a weak joint which might alter the frequency trend. Since the proposed method uses a random location for multiple damages, the random change of the damage size and position might have an impact on the frequency shift.

Figure 10 provides the calculated main effects of different variables for different damage conditions. Figure 10a–c shows that damage severity (λ) has the higher impact on the frequency shift comparing to the damage sizes (both the mean and standard deviation value). The  λ value is co-related with the change of the cross-sectional area. So, the result depicts that any damage extent will highly impact the frequency shift. In mode 1, the effect of the µ value is three times higher than α, whereas, in mode 2 and 3, it shows the opposite trends. In these modes, the SD is less effective than the mean values. Having a low SD (µ) value means damages are more closely clustered and a high SD (µ) value means damages are more scattered from the mean value. Here, the effect of µ is three and 1.2 times less than α, respectively, in mode 2 and 3 (Figure 10b,c), which means the standard deviation of damage sizes does not affect the frequency change more than the mean values in these modes. Thus, for the range of variables considered in the analysis, the damage severity is the most impactful on the frequency shift while the mean and SD of damage size is also important but has less effect than λ.

Table 7, Table 8 and Table 9 show the comparison of relative error among the FE model, ANN model, and statistical analysis for different modes considering different damage sizes and severity. The comparison is developed using the new damage conditions. It is observed that the relative error between the ANN model and statistical solution to predict the natural frequency of the damaged beam is 0.22–3.68% in mode 1, 0.8–3% in mode 2, and 0.89–3.26% in mode 3. The beam geometry, material properties along with its random damage size and location might affect its vibration characteristics. Considering this, the error result indicates that the developed ANN model can predict the frequency satisfactorily. Among these three models, the ANN model gives the most convincing output with less computational error, time, and complexity.

The above-mentioned result and discussion show that among three variables (damage severity, damage mean size, and standard deviation), damage severity has the most impacts on the frequency change. The proposed ANN model is a good alternative to the FE model since minimal error (average maximum 22% to minimum 6% in each mode) is obtained for any damage case. This model can successfully predict the natural frequency for beams with multiple damages at a random location and can save modeling analysis time that is required for the FE technique.

## 5. Conclusions

The free vibration of a damaged cantilever beam affects its modal parameters like its frequency and mode shape. This study shows that damage severity highly impacts the frequency shift, rather than the damage size. In this paper, the proposed neural network model is an improvement in the field of vibration-based damage identification by providing a reasonable prediction of the natural frequencies of a cantilever beam with multiple damages at a random location. It shows that with the increase of the damage size and severity, the beam frequency will decrease since damage has impacts on its cross-sectional area and stiffness properties. Moreover, the frequency of the higher mode is always greater than its immediate lower mode for any damage condition. The frequency results provided by the developed model agree well with the FE result. Therefore, the study can efficiently measure the frequency changes in terms of damage severity, damage size, and random location, which contributes as a helpful tool in a structural health monitoring system to predict the damage state with a minimal computational burden.

## Figures and Tables

**Figure 1 sensors-23-07867-f001:**
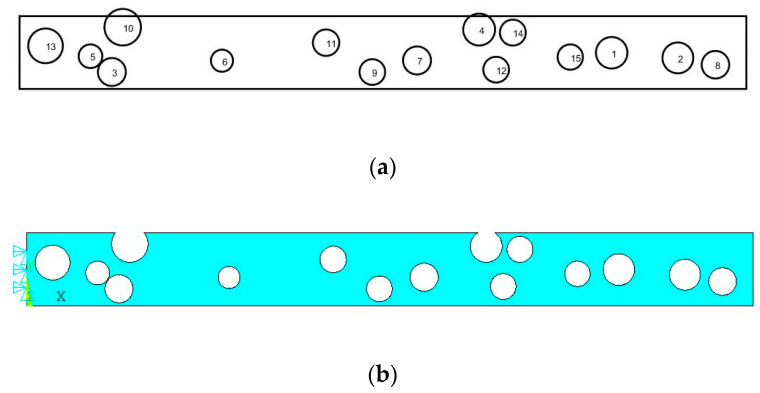
An example of the beam with random multiple damages by using the Monte Carlo method in MATLAB (**a**), and the damaged beam model in APDL (**b**).

**Figure 2 sensors-23-07867-f002:**
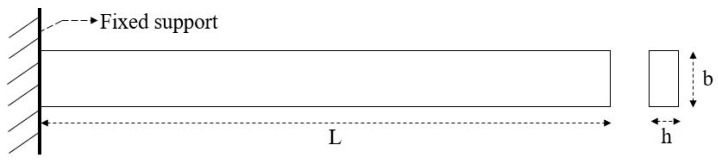
Cantilever undamaged beam fixed at one end.

**Figure 3 sensors-23-07867-f003:**
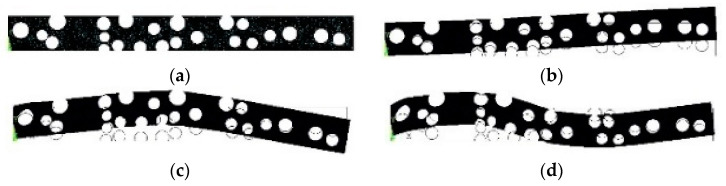
Simulation result of cantilever beam using the APDL script: undeformed beam (**a**); deformed beam in mode 1 (**b**), mode 2 (**c**), and mode 3 (**d**) after free vibration.

**Figure 4 sensors-23-07867-f004:**
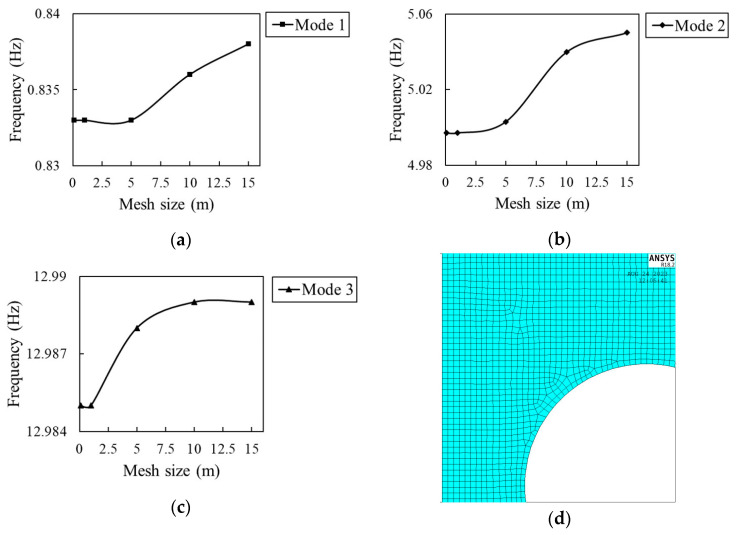
Mesh sensitivity result: (**a**) mode 1, (**b**) mode 2, and (**c**) mode 3. (**d**) APDL meshed elements near the damaged area.

**Figure 5 sensors-23-07867-f005:**
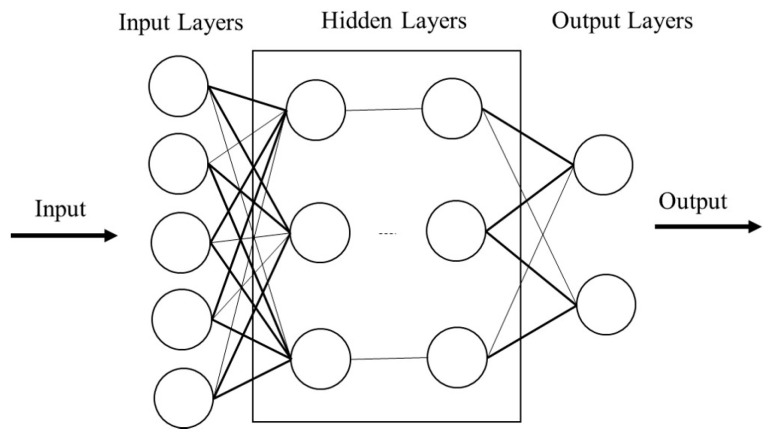
Typical NN structure.

**Figure 6 sensors-23-07867-f006:**
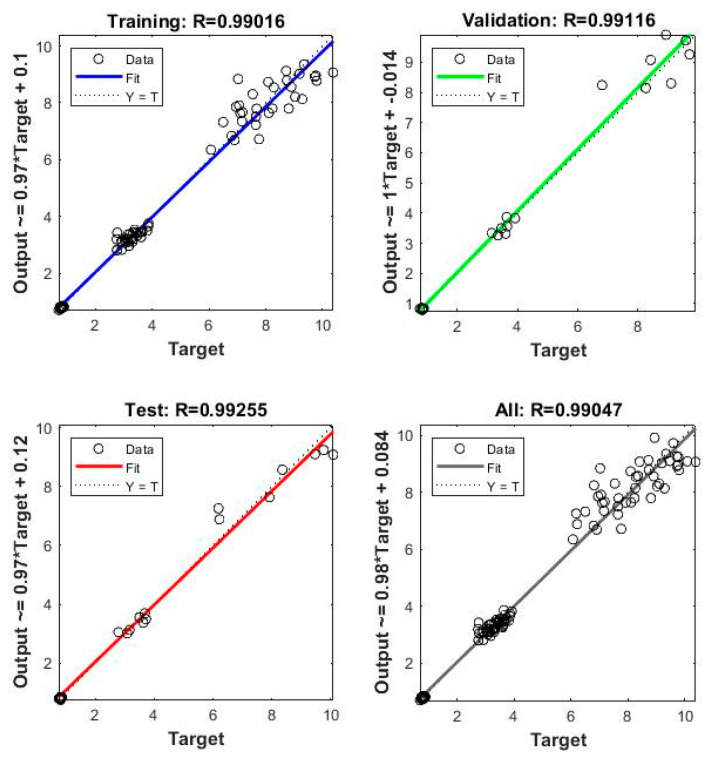
ANN model performance.

**Figure 7 sensors-23-07867-f007:**
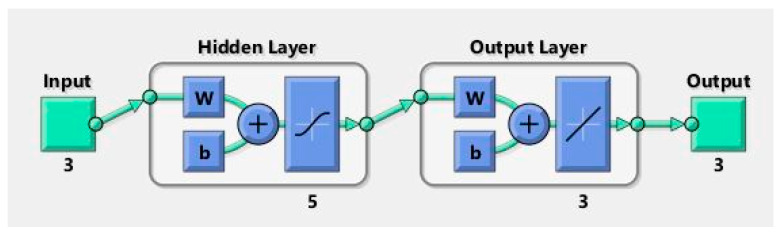
ANN architecture (w = weight, b = bias).

**Figure 8 sensors-23-07867-f008:**
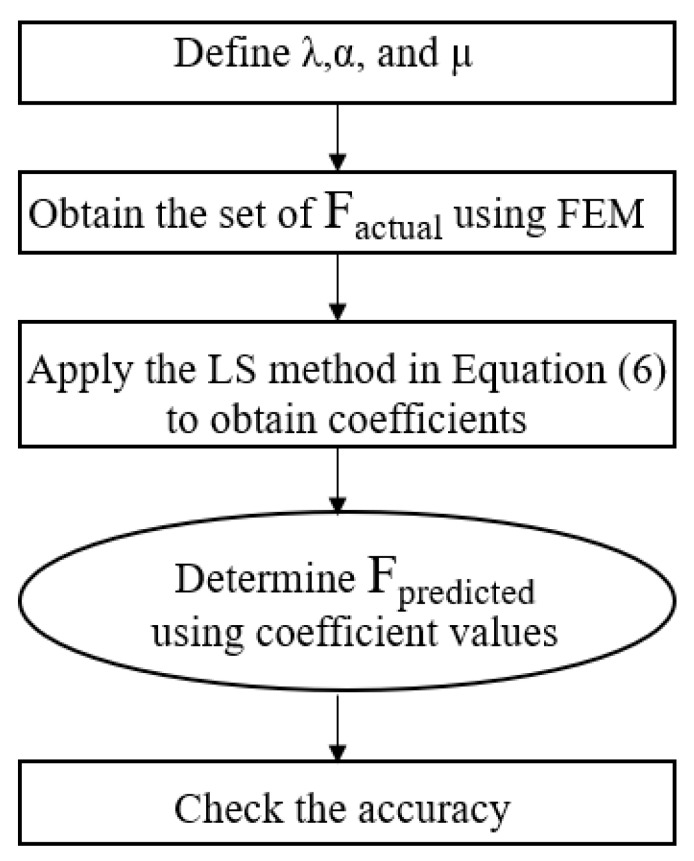
Flow chart to calculate predicted natural frequencies with the statistical approach.

**Figure 9 sensors-23-07867-f009:**
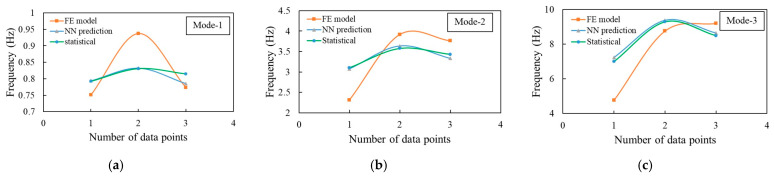
Comparison of FE Frequency with ANN model and statistical formula: (**a**) mode 1, (**b**) mode 2, and (**c**) mode 3.

**Figure 10 sensors-23-07867-f010:**
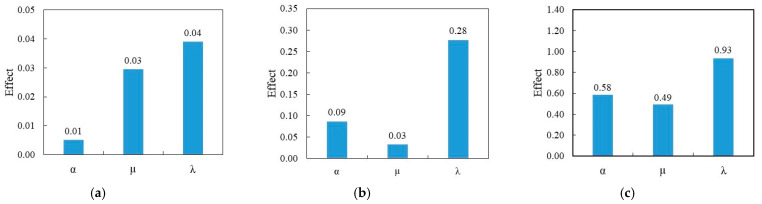
Effect of different variables on the frequency shift. Bar chart of the effect in (**a**) mode 1, (**b**) mode 2, and (**c**) mode 3.

**Table 1 sensors-23-07867-t001:** Material properties.

Variables	Properties
Beam material	Steel
Youngs modulus, E	2.1 × 10^11^ Pa
Density, ρ	7800 kg/m^3^
Poisson’s ratio for steel	0.28

**Table 2 sensors-23-07867-t002:** Levels of parameters used in the data sensitivity analysis.

Variables	Low (−)	High (+)
α	2 m	2.5 m
µ	0.1 m	0.2 m
λ	10%	20%

**Table 3 sensors-23-07867-t003:** Frequency result of undamaged beam.

Mode No.	THEORY (Hz)	FEM (Hz)
1	0.8381	0.838
2	5.253	5.0503
3	14.822	12.989

**Table 4 sensors-23-07867-t004:** Predicted frequency of the FE model, ANN, and statistical analysis for the new dataset (Mode 1).

α (m)	µ (m)	λ	FE Model (F′, Hz)	ANN (F″, Hz)	Statistical Regression (F‴, Hz)
3	30%	30%	0.7508	0.7940	0.7923
2.5	20%	20%	0.9472	0.8324	0.8307
2	10%	10%	0.7736	0.7958	0.8147

**Table 5 sensors-23-07867-t005:** Predicted frequency of the FE model, ANN, and statistical analysis for the new dataset (Mode 2).

α (m)	µ (m)	λ	FE Model (F′, Hz)	ANN (F″, Hz)	Statistical Regression (F‴, Hz)
3	0.30	30%	2.3133	3.0803	3.1058
2.5	0.20	20%	3.9156	3.6380	3.5766
2	0.10	10%	3.7615	3.3360	3.4325

**Table 6 sensors-23-07867-t006:** Predicted frequency of the FE model, ANN, and statistical analysis for the new dataset (Mode 3).

α (m)	µ (m)	λ	FE Model (F′, Hz)	ANN (F″, Hz)	Statistical Regression (F‴, Hz)
3	0.30	30%	4.7705	7.2338	6.9982
2.5	0.20	20%	8.7671	9.3728	9.2894
2	0.10	10%	9.2070	8.6486	8.4918

**Table 7 sensors-23-07867-t007:** Relative error comparison in Mode 1.

α (m)	µ (m)	λ	Relative Error (RE, %)		
			FE Model vs. ANN	FE Model vs. Statistical Regression	ANN vs. Statistical Regression
3	0.30	30%	5.76	5.53	0.22
2.5	0.20	20%	11.18	11.37	0.21
2	0.10	10%	1.57	5.31	3.68

**Table 8 sensors-23-07867-t008:** Relative error comparison in Mode 2.

α (m)	µ (m)	λ	Relative Error (RE, %)		
			FE Model vs. ANN	FE Model vs. Statistical Regression	ANN vs. Statistical Regression
3	0.30	30%	33.16	34.26	0.83
2.5	0.20	20%	7.09	8.66	1.69
2	0.10	10%	11.31	8.75	2.90

**Table 9 sensors-23-07867-t009:** Relative error comparison in Mode 3.

α (m)	µ (m)	λ	Relative Error (RE, %)		
			FE Model vs. ANN	FE Model vs. Statistical Regression	ANN vs. Statistical Regression
3	0.30	30%	51.64	46.70	3.26
2.5	0.20	20%	6.91	5.96	0.89
2	0.10	10%	6.17	7.77	1.70

## Data Availability

The data presented in this study are available in Figshare at https://doi.org/10.6084/m9.figshare.24130101 (accessed on 11 September 2023).

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
