# Peer review of "A Neural Network Approach to Estimate the Frequency of a Cantilever Beam with Random Multiple Damages"

_sensors, 2023, doi:10.3390/s23187867_

Round 1

Reviewer 1 Report

The paper presents an approach for detecting structural damage in beams using artificial neural networks (ANNs) and natural frequency data. The Monte Carlo simulation is employed to generate various damage cases, and the ANN is trained to learn the complex relationships between damage parameters and natural frequency shifts. The paper has not been well-written, especially “Methodology” section. The results and conclusions of this research are very limited. The following comment should be considered when revising the manuscript.

·       The contribution of the research is not clear. This should be declared in “Abstract” and “Introduction”.

·       When abbreviating a term, use the full term the first time you use it in the text. Fore instant “APDL”.

·       The references cited in this work are old. Please substitute them with the research papers published recently (after 2017). The following papers can be referred.

·       Zhang, C., Mousavi, A.A., Masri, S.F., Gholipour, G., Yan, K., Li, X. Vibration feature extraction using signal processing techniques for structural health monitoring: A review (2022) Mechanical Systems and Signal Processing, 177, DOI: 10.1016/j.ymssp.2022.109175

·       Mousavi, A.A., Zhang, C., Masri, S.F., Gholipour, G. Damage detection and localization of a steel truss bridge model subjected to impact and white noise excitations using empirical wavelet transform neural network approach (2021) Measurement: Journal of the International Measurement Confederation, 185, DOI: 10.1016/j.measurement.2021.110060

·       Mousavi, A.A., Zhang, C., Masri, S.F., Gholipour, G. Structural damage localization and quantification based on a CEEMDAN hilbert transform neural network approach: A model steel truss bridge case study (2020) Sensors (Switzerland), 20 (5), DOI: 10.3390/s20051271

·       The methodology section is not well-written. In Page 3, the concept cited in “Lines 108-114” should be described more in detail. It is not clear that how the damage severities are assumed to be between 10% and 30%.

·       Figure 1 is not suitable. It is recommended to present the FE model of the beam developed in ANSYS along with sufficient information. Please improve the presentation of all the figures throughout the manuscript.

·       More information of the FE model of the beam should be presented. For instance, how are cross-section details, the meshing, and boundary conditions visually?

·       Did you consider any steel reinforcements for the beam? Please justify the selection of case study.

·       What is the unit for the mesh size considered (i.e., 0.1)?

·       How did you select a mesh size of 0.1? Did the authors any mesh convergence tests? If yes, please present the results.

·       Page 4, Lines 148-150: Table 1 does not provide the information about boundary condition. This table list the material properties.

·       Please justify the sufficiency of the hidden layers considered for this study.

·       Could you provide more details about the architecture of the neural network used for predicting natural frequencies? What were the considerations behind choosing a specific network architecture, including the number of layers and neurons?

·       How was the dataset for training and validation generated? What is the size and diversity of the dataset in terms of damage scenarios, beam geometries, and material properties?

·       Were any measures taken to ensure the dataset covers a representative range of damage types and severities that can occur in real-world scenarios?

·       While the Monte Carlo method allows for various damage scenarios, how accurately does it represent real-world damage evolution and distribution? What assumptions or simplifications were made during the damage simulation process?

·       How do the artificially generated damage scenarios compare to actual damage patterns observed in structural components subjected to operational loads and environmental conditions?

·       Can the trained neural network provide insights into which specific damage parameters or combinations thereof have the most significant influence on the predicted natural frequencies?

·       How sensitive is the neural network's performance to noise in the input data, such as measurement errors in natural frequency extraction or sensor noise?

·       Have you investigated the effects of uncertainties in material properties, boundary conditions, or other modeling assumptions on the accuracy of damage prediction?

·       How would the proposed method integrate with real-world structural health monitoring systems? What are the challenges associated with deploying this approach in practical scenarios with limited sensor data or operational variability?

·       Could you elaborate on the limitations of your study? What are the specific scenarios or types of damage for which your method might struggle to provide accurate predictions?

NA

Reviewer 2 Report

The paper introduces an Artificial Neural Network (ANN) model designed to predict the natural frequencies of a damaged cantilever beam, a crucial element in vibration-based damage identification. Leveraging the Monte Carlo method for varied damage simulations and a MATLAB script for efficient Finite Element (FE) interfacing, the ANN model's predictions consistently aligned with both FE results and a 3rd order polynomial linear regression model.

Major comments:

1. The paper does not elaborate on the exact architecture and parameters of the ANN model. Details about layers, neurons, activation functions, or training methodology could have provided greater insights into the model's design.

2. While the paper does a good job comparing the ANN model with the Finite Element method and a statistical solution, exploring comparisons with other existing methodologies or models in the domain might have given a more comprehensive understanding of where the proposed model stands.

3. The training dataset for the neural network was based on the FE model. Any inherent inaccuracies or limitations in the FE model would propagate into the ANN model, potentially affecting its prediction capability.

4. The paper notes a peculiar behavior where the frequency shift for 20% damage sometimes surpasses those for both 10% and 30% damages, attributed to damage location and size. This phenomenon could be explored further to understand its implications and conditions.

5. The paper employs a 3rd order polynomial equation for validation. Given the complex nature of the problem, multiple validation methodologies could have provided a more robust validation of the ANN model.

6. The paper mainly focuses on random damage locations with varying sizes and severities. An exploration of systematic damages or considering other external factors might offer a more holistic solution.

7. The paper does not seem to discuss potential limitations or challenges of the proposed ANN model explicitly. Understanding the scenarios where the model might not be as effective or challenges that could arise would have been valuable.

8. While the paper demonstrates the model's potential in a controlled environment, recommendations or guidelines for real-world applications, considering noise, varying environmental factors, or real-time processing demands, might have further bridged the gap between theory and practical application.

9. Training an ANN model using data generated from a FE model can lead to overfitting, especially if the ANN is particularly complex. The paper doesn't discuss steps taken to mitigate overfitting or any cross-validation results.

10. The paper describes multiple damages with varying size and severity but doesn't delve deep into defining what constitutes different levels of severity, potentially leaving ambiguities in interpretation.

11. Most cited papers are too stale.

1. Some sentences are quite lengthy and could benefit from restructuring to improve clarity.

2. Certain phrases and concepts are repeated multiple times in close succession, which can make the reading slightly redundant.

3. There are a few instances where terms or references might not be consistently used. For example, "Eq (1)" and "Eq.(2)" - it would be better to choose a single format and stick to it throughout the paper.

Reviewer 3 Report

Comments to the authors:

1. Kindly avoid the term 'we' in the article.

2. Describe the novelty of the article.

3. More than 50% of the references are old. Advised to refer to recent literature (2020-2023)

4. How does the neural network-based approach for natural frequency prediction in a cantilever beam with random multiple damages compare to the traditional finite element method in terms of accuracy and computational efficiency?

5. Kindly explore the hyper-tuning parameters of ANN.

6. Suggested to deploy ANN with different architectures and transfer functions to arrive at the best results.

7. Suggested to compare with other ML algorithms to ensure the accuracy of ANN

8. Could you elaborate on the advantages and limitations of using the neural network solution, particularly in terms of accuracy, computational efficiency, and applicability to various damage scenarios?

9. Additionally, how significant is the contribution of the curve-fitting equation in validating the accuracy of predicted natural frequencies, and what insights can be gained from the frequency results in terms of the relationship between damage parameters and frequency alterations?

10. How does the proposed work significantly contribute towards SHM in the industry?

11. Conclusion has to be updated with research findings.

Moderate editing of the English language required

Round 2

Reviewer 2 Report

Thanks for addressing my comments.

Reviewer 3 Report

Congrats to the authors.